# Information Bottleneck: Exact Analysis of (Quantized) Neural Networks

**Stephan S. Lorenzen, Christian Igel & Mads Nielsen**
Department of Computer Science
University of Copenhagen
`{lorenzen,igel,madsn}@di.ku.dk`

## Abstract

The information bottleneck (IB) principle has been suggested as a way to analyze deep neural networks. The learning dynamics are studied by inspecting the mutual information (MI) between the hidden layers and the input and output. Notably, separate fitting and compression phases during training have been reported. This led to some controversy including claims that the observations are not reproducible and strongly dependent on the type of activation function used as well as on the way the MI is estimated. Our study confirms that different ways of binning when computing the MI lead to qualitatively different results, either supporting or refusing IB conjectures. To resolve the controversy, we study the IB principle in settings where MI is non-trivial and can be computed exactly. We monitor the dynamics of quantized neural networks, that is, we discretize the whole deep learning system so that no approximation is required when computing the MI. This allows us to quantify the information flow without measurement errors. In this setting, we observed a fitting phase for all layers and a compression phase for the output layer in all experiments; the compression in the hidden layers was dependent on the type of activation function. Our study shows that the initial IB results were not artifacts of binning when computing the MI. However, the critical claim that the compression phase may not be observed for some networks also holds true.

## 1 Introduction

Improving our theoretical understanding of why over-parameterized deep neural networks generalize well is arguably one of main problems in current machine learning research (Poggio et al., 2020). Tishby & Zaslavsky (2015) suggested to analyze deep neural networks based on their *Information Bottleneck (IB)* concept, which is built on measurements of *mutual information* (MI) between the activations of hidden layers and the input and target (Tishby et al., 1999), for an overview see Geiger (2020). Shwartz-Ziv & Tishby (2017) empirically studied the IB principle applied to neural networks and made several qualitative observations about the training process; especially, they observed a *fitting* phase and a *compression* phase. The latter information-theoretic compression is conjectured to be a reason for good generalization performance and has widely been considered in the literature (Abrol & Tanner, 2020; Balda et al., 2018; 2019; Chelombiev et al., 2019; Cheng et al., 2019; Darlow & Storkey, 2020; Elad et al., 2019; Fang et al., 2018; Gabrié et al., 2019; Goldfeld et al., 2019; Jónsson et al., 2020; Kirsch et al., 2020; Tang Nguyen & Choi, 2019; Noshad et al., 2019; Schiemer & Ye, 2020; Shwartz-Ziv & Alemi, 2020; Wickstrøm et al., 2019; Yu et al., 2020). The work and conclusions by Shwartz-Ziv & Tishby (2017) received a lot of critique, with the generality of their claims being doubted; especially Saxe et al. (2018) argued that the results by Shwartz-Ziv & Tishby do not generalize to networks using a different activation function. Their critique was again refuted by the original authors with counter-claims about incorrect estimation of the MI, highlighting an issue with the approximation of MI in both studies.

Our goal is to verify the claims by Shwartz-Ziv & Tishby and the critique by Saxe et al. in a setting where the MI can be computed exactly. These studies consider neural networks as theoretical entities working with infinite precision, which makes computation of the information theoretic quantities problematic (for a detailed discussion we refer to Geiger, 2020, see also Section 3). Assuming

continuous input distributions, a deterministic network using any of the standard activation functions (e.g., RELU, TANH) can be shown to have infinite MI (Amjad & Geiger, 2019). If an empirical input distribution defined by a data set $\mathcal{D}$ is considered (as it is the case in many of the previous studies), then randomly-initialized deterministic neural networks with invertible activation functions will most likely result in trivial measurements of MI in the sense that the MI is finite but always maximal, that is, equal to $\log |\mathcal{D}|$ (Goldfeld et al., 2019; Amjad & Geiger, 2019). In order to obtain non-trivial measurements of MI, real-valued activations are usually discretized by *binning* the values, throwing away information in the process. The resulting estimated MI can be shown to be highly dependent on this binning, we refer to Geiger (2020) for a detailed discussion. Instead of approximating the MI in this fashion, we take advantage of the fact that modern computers – and thus neural networks – are discrete in the sense that a floating point value can typically take at most $2^{32}$ different values. Because 32-bit precision networks may still be too precise to observe compression (i.e., information loss), we apply *quantization* to the neural network system to an extent that we can compute informative quantities, that is, we amplify the effect of the information loss due to the discrete computations in the neural network. One may argue that we just moved the place where the discretization is applied. This is true, but leads to a fundamental difference: previous studies applying the discretization post-hoc rely on the in general false assumption that the binned MI approximates the continuous MI well – and thus introduce measurement errors, which may occlude certain phenomena and/or lead to artifactual observations. In contrast, our computations reflect the true information flow in a network during training. Our study confirms that estimation of MI by binning may lead to strong artifacts in IB analyses and shows that:

- Both fitting and compression phases occur in the output SOFTMAX layer.

- For the hidden layers, the fitting phase occurs for both TANH and RELU activations.

- When using TANH in the hidden layers, compression is only observed in the last hidden layer.

- When using RELU, we did not observe compression in the hidden layers.

- Even when applying low precision quantization, more complex networks with many neurons in each layer are observed to be too expressive to exhibit compression, as no information is lost.

- Our setting excludes that the MI approximation is the reason for these different IB dynamics.

The next section introduces the IB concept with a focus on its application to neural networks including the critique and controversy as well as related work. Section 3 discusses issues relating to the estimation of MI, and the idea behind our contribution. Section 4 presents our experiments, results and discussion before we conclude in Section 5.

## 2 THE INFORMATION BOTTLENECK

**Preliminaries.** Given a continuous *random variable (r.v.)* $X$ with density function $p(x)$ and support $\mathcal{X}$, the continuous entropy $\mathrm{H}(X)$ of $X$ is a measure of the *uncertainty* associated with $X$ and is given by $\mathrm{H}(X) = -\int_{\mathcal{X}} p(x) \log p(x) \mathrm{d}x$. Given two r.v.s $X$ and $Y$ with density functions $p(x)$ and $q(y)$ and supports $\mathcal{X}$ and $\mathcal{Y}$, the mutual information $\mathrm{I}(X;Y)$ of $X$ and $Y$ is a measure of the mutual "knowledge" between the two variables. The symmetric $\mathrm{I}(X;Y)$ is given by $\mathrm{I}(X;Y) = \int_{\mathcal{Y}} \int_{\mathcal{X}} p(x,y) \log \frac{p(x,y)}{p(x)p(y)} \mathrm{d}x\mathrm{d}y$. In many cases it is impossible to compute the continuous entropy and MI for continuous r.v.s exactly, due to limited samples or computational limits, or because it may not be finite (Geiger, 2020). Instead, we often estimate the quantities by their discrete counterparts. When $X$ is a discrete r.v., we consider the *Shannon entropy* $\mathrm{H}(X) = -\sum P(x) \log P(x)$. Correspondingly, the mutual information $\mathrm{I}(X;Y)$ of two discrete r.v.s $X,Y$, is given by $\mathrm{I}(X;Y) = \sum_{x,y} P(x,y) \log \frac{P(x,y)}{P(x)P(y)}$. We have the following useful identity for both the continuous and discrete MI:

$$\mathrm{I}(X;Y) = \mathrm{H}(X) - \mathrm{H}(X|Y) \ , \tag{1}$$

where $\mathrm{H}(X|Y)$ is the conditional entropy of $X$ given $Y$.

**IB Definition.** The IB method was proposed by Tishby et al. (1999). It is an information theoretic framework for extracting relevant components of an *input* r.v. $X$ with respect to an *output* r.v. $Y$. These relevant components are found by "squeezing" the information from $X$ through a *bottleneck*, in the form of an r.v. $T$. In other words, $T$ is a compression of $X$. The idea generalizes *rate distortion theory*, in which we wish to compress $X$, obtaining $T$, such that $\mathrm{I}(X;T)$ is maximized subject to a constraint on the expected distortion $d(x,t)$ wrt. the joint distribution $p(x,t)$ (Tishby et al., 1999). In the IB framework, the distortion measure $d$ is replaced by the negative loss in MI between $T$ and the output $Y$, $\mathrm{I}(T;Y)$. Both IB and rate distortion are lossy compression schemes.[1]

The *data processing inequality (DPI)* $\mathrm{I}(Y;X) \geq \mathrm{I}(Y;T)$ holds for the IB; that is, the bottleneck r.v. cannot contain more information about the label than the input.

One drawback of the information bottleneck method is the dependence on the joint distribution, $p(x,y)$, which is generally not known. Shamir et al. (2010) addressed this issue and showed that the MI, the main ingredient in the method, can be estimated reliably with fewer samples than required for estimating the true joint distribution. As common in the IB literature, whenever we discuss the MI computed on a finite data set $\mathcal{D}$, we assume that $p(x,y)$ corresponds to the empirical distribution defined by $\mathcal{D}$, which is true for the experiments in Section 4.1. In practice, the assumption has to be relaxed to the data being drawn i.i.d. from $p(x,y)$. However, any uncertainty resulting from the finite sample estimation in the latter case is not considered in our discussions.

**IB In Deep Learning.** Tishby & Zaslavsky (2015) applied the IB concept to neural networks. They view the layers of a *deep neural network (DNN)* as consecutive compressions of the input. They consider the Markov chain

$$Y \to X \to T_1 \to T_2 \to ... \to T_L = \hat{Y} ,$$

where $T_i$ denotes the $i$'th hidden layer of the $L$-layer network and $T_L = \hat{Y}$ denotes the output of the network. Again, the bottleneck must satisfy the DPI:

$$\mathrm{I}(Y;X) \geq \mathrm{I}(Y;T_1) \geq \mathrm{I}(Y;T_2) \geq ... \geq \mathrm{I}\left(Y;\hat{Y}\right) , \tag{2}$$

$$\mathrm{I}(X;X) \geq \mathrm{I}(X;T_1) \geq \mathrm{I}(X;T_2) \geq ... \geq \mathrm{I}\left(X;\hat{Y}\right) . \tag{3}$$

Estimating the MI of continuous variables is difficult (Alemi et al., 2017), as evident from the many different methods proposed (Kraskov et al., 2004; Kolchinsky & Tracey, 2017; Noshad et al., 2019). In the discrete case, $\mathrm{I}(X;T)$ and $\mathrm{I}(T;Y)$ can be computed as

$$\mathrm{I}(X;T) = \mathrm{H}(T) - \mathrm{H}(T|X) = \mathrm{H}(T) , \tag{4}$$

$$\mathrm{I}(T;Y) = \mathrm{I}(Y;T) = \mathrm{H}(T) - \mathrm{H}(T|Y) , \tag{5}$$

following from (1) and using in (4) the assumption that $T$ is a deterministic function of $X$. However, for deterministic neural networks the continuous entropies may not be finite (Goldfeld et al., 2019; Saxe et al., 2018; Amjad & Geiger, 2019).

Shwartz-Ziv & Tishby (2017) estimate the MI via (4) and (5) by discretizing $T$ and then computing the discrete entropy. They trained a network (shown in Figure 1a) on a balanced synthetic data set consisting of 12-bit binary inputs and binary labels. The network was trained for a fixed number of epochs while training and test error were observed. For every epoch and every layer $T$, The discretization is done by *binning* of $T$: Given upper and lower bounds $b_u, b_l$, and $m \in \mathbb{N}$, we let $B : \mathbb{R} \to [m]$ denote the binning operation, that maps $x \in [b_l, b_u]$ to the index of the corresponding bin from the set of $m$ uniformly distributed bins in $[b_l, b_u]$. Overloading the notation, we apply $B$ directly to a vector in $\mathbb{R}^d$ in order to obtain the resulting vector in $[m]^d$ of bin indices. Using discretized $T' = B(T)$, $\mathrm{I}(X;T')$ and $\mathrm{I}(T';Y)$ are then computed directly by (4) and (5), using estimates of $P(T')$, $P(Y)$ and $P(T'|Y)$ over all samples $\mathcal{D}$ of $X$.

Shwartz-Ziv & Tishby used the TANH activation function for the hidden layers, with $b_l = -1, b_u = 1$ ($b_l = 0$ for the output SOFTMAX layer) and $m = 30$ bins. The estimated $\mathrm{I}(X;T)$ and $\mathrm{I}(T;Y)$ are plotted in the *information plane*, providing a visual representation of the information flow in

---

[1]For IB, finding the optimal representation $T$ can be formulated as the minimization of the Lagrangian $\mathrm{I}(X;T) - \beta\mathrm{I}(T;Y)$ subject to the Markov chain $Y \to X \to T$ and $\beta \in \mathbb{R}^+$ (Tishby & Zaslavsky, 2015).

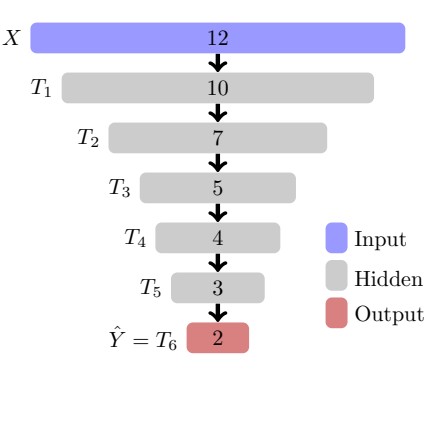

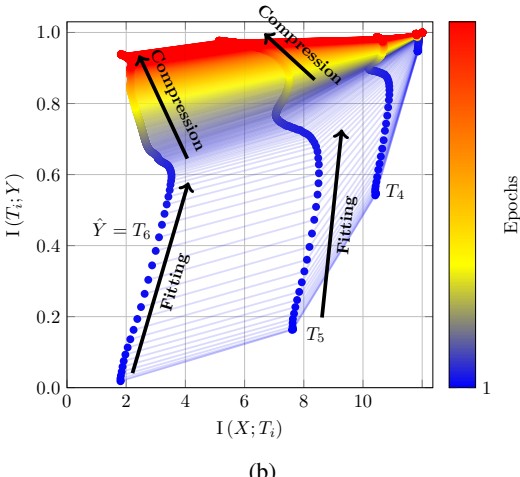

(a)

(b)

Figure 1: (a) The network used in the standard setting, with the number of neurons in each fully connected layer. The output layer activation function is SOFTMAX, while either TANH or RELU is used for the hidden layers. (b) An example of an information plane for a 6 layer network, with the observed phases marked. Only the last three layers are distinguishable.

the network during training (see example in Figure 1b). Based on the obtained results, Shwartz-Ziv & Tishby (2017) make several observations; one notable observation is the occurrence of two phases: an *empirical risk minimization* phase and a *compression* phase. The first phase, also referred to as *fitting phase*, is characterized by increasing $I(T; Y)$ related to a decreasing loss. The subsequent compression phase is characterized by decreasing $I(X; T)$, and it has been argued that this compression leads to better generalization.

**Critique and Controversy.** The work by Tishby & Zaslavsky (2015) and Shwartz-Ziv & Tishby (2017) has jump-started an increasing interest in the IB method for deep learning, with several papers investigating or extending their contributions, see the review by Geiger (2020). However, as mentioned, their work also received criticism. In particular, the compression phase as a general phenomenon has been called into question. Saxe et al. (2018) published a paper refuting several of the claims made by Shwartz-Ziv & Tishby (2017). They based their criticism on a replication of the experiment done by Shwartz-Ziv & Tishby (2017) where they replaced the bounded TANH activation function with the unbounded RELU activation function. When discretizing, they used the maximum activation observed across all epochs for the upper binning bound $b_u$. The article claims that the two phases observed by Shwartz-Ziv & Tishby occurred because the activations computed using TANH saturate close to the boundaries $-1$ and $1$. The claim is supported by experiments using RELU activations and $m = 100$ bins, in which the two phases are not observed.

The critique paper by Saxe et al. was published at ICLR 2018, but started a discussion already during the review process the previous year, when Shwartz-Ziv & Tishby defended their paper in the online discussion forum OpenReview.net[2], posting a response titled "Data falsifying the claims of this ICLR submission all together" (Saxe et al., 2017). The response specifically states that "The authors don't know how to estimate mutual information correctly", referring to Saxe et al. (2018), and goes on to provide an example with a network using RELU activations, which does indeed exhibit the two phases. In response, Saxe et al. performed further experiments using different estimators for MI: a state-of-the-art non-parametric KDE approach (Kolchinsky & Tracey, 2017) and a $k$-NN based estimator (Kraskov et al., 2004). The authors still did not observe the two phases claimed by Shwartz-Ziv & Tishby.

Following the discussion on OpenReview.net, several other papers have also commented on the controversy surrounding the information bottleneck. Noshad et al. (2019) presented a new MI estimator EDGE, based on dependency graphs, and tested it on the specific counter example using

---

[2]https://openreview.net/

RELU activations as suggested by Saxe et al. (2018), and they observed the two phases. Table I in the review by Geiger (2020) provides a nice overview over empirical IB studies and if the compression phase was observed (Darlow & Storkey, 2020; Jónsson et al., 2020; Kirsch et al., 2020; Noshad et al., 2019; Raj et al., 2020; Shwartz-Ziv & Tishby, 2017) or not (Abrol & Tanner, 2020; Balda et al., 2018; 2019; Tang Nguyen & Choi, 2019; Shwartz-Ziv & Alemi, 2020; Yu et al., 2020) or the results were mixed (Chelombiev et al., 2019; Cheng et al., 2019; Elad et al., 2019; Fang et al., 2018; Gabrié et al., 2019; Goldfeld et al., 2019; Saxe et al., 2018; Schiemer & Ye, 2020; Wickstrøm et al., 2019). In conclusion, an important part of the controversy surrounding the IB hinges on the estimation of the information-theoretic quantities – this issue has to be solved before researching the information flow.

**Related Work.** The effect of estimating MI by binning has been investigated before, we again refer to Geiger (2020) for a good overview and discussion. Shwartz-Ziv & Alemi (2020) consider infinite ensembles of infinitely-wide networks, which renders MI computation feasible, but do not observe a compression phase. Chelombiev et al. (2019) applies adaptive binning, which, while less prone to issues caused by having the "wrong" number of bins, is still an estimation and thus also suffers from the same problems. Goldfeld et al. (2019) explores IB analysis by use of stochastic neural networks which allow them to show that the compression phase in these noisy networks occurs due to clustering of the hidden representations. While theoretically interesting, the stochastic neural networks are still qualitatively different from deterministic ones and thus not directly applicable in practice. Raj et al. (2020) conducted an IB analysis of binary networks, where both the activations and weights can only be $\pm 1$, which allows for exact computation of MI. The binary networks are significantly different from the networks used in the original studies, whereas applying the IB analysis to quantized versions of networks from these studies allows for a more direct comparison. At a first glance, the work by Raj et al. (2020) could be viewed as taking our approach to the extreme. However, the computations and training dynamics of the binary networks are qualitatively different from our study and the original IB work. For example, the binary networks require modifications for gradient estimation (e.g., Raj et al. consider *straight-through-estimator*, *approximate sign*, and *swish sign*).

## 3 ESTIMATING ENTROPY AND MUTUAL INFORMATION IN NEURAL NETWORKS

The IB principle is defined for continuous and discrete input and output, however continuous MI in neural networks is in general infeasible to compute, difficult to estimate, and conceptually problematic.

### 3.1 BINNING AND LOW COMPRESSION IN PRECISE NETWORKS

Consider for a moment a deterministic neural network on a computer with infinite precision. Assuming a constant number of training and test patterns, a constant number of update steps, invertible activation functions in the hidden layers and pairwise different initial weights (e.g., randomly initialized). Given any two different input patterns, two neurons will have different activations with probability one (Goldfeld et al., 2019; Geiger, 2020). Thus, on any input, a hidden layer $T$ will be unique, meaning that, when analysing data set $\mathcal{D}$, all observed values of $T$ will be equally likely, and the entropy $\mathrm{H}(T)$ will be trivial in the sense that it is maximum, that is, $\log |\mathcal{D}|$.

However, when applying the binning $T' = B(T)$, information is dropped, as $B$ is surjective and several different continuous states may map to the same discrete state; for $m$ bins and $d_T$ neurons in $T$ the number of different states of $T'$ is $m^{d_T}$, and thus more inputs from $\mathcal{D}$ will map to the same state. As a consequence, the estimated entropy is decreased. Furthermore, the rate of decrease follows the number of bins; the smaller the number of bins $m$, the smaller the estimated $\mathrm{H}(T)$ (as more values will map to the same state). The entropy $\mathrm{H}(T')$, and therefore $\mathrm{I}(X; T')$ in deterministic networks, is upper bounded by $d_T \log m$. Thus, changing the number of bins used in the estimation changes the results. We also refer to Geiger (2020) for a more detailed discussion of this phenomenon[3].

Computations in digital computers are not infinitely precise, a 32-bit floating point variable, for instance, can take at most $2^{32}$ different values. In theory, this means that we can actually consider the states of a neural network discrete and compute the exact MI directly. In practice, we expect the precision to be high enough, that – when using invertible activation functions – the entropy of the

---

[3]Note that Geiger, in contrast to our terminology, uses *quantizer* or $Q$ to denote post-hoc binning.

layers will be largely uninformative, as almost no information loss/compression will occur. However, using low precision *quantized* neural networks, we can compute the exact MI in the network without $H(T)$ being trivial – what we regard as the ideal setting to study IB conjectures.

## 3.2 Exact Mutual Information in Quantized Networks

Quantization of neural networks was originally proposed as a means to create more efficient networks, space-wise and in terms of inference running time, for instance to make them available on mobile devices (Jacob et al., 2018; Hubara et al., 2017). The idea is to use low precision $k$-bit weights (typically excluding biases) and activations, in order to reduce storage and running time. For our use case, we are mainly interested in quantized activations, as they contain the information that is moving through the network.

The naive approach is to quantize after training (*post-training quantization*). In terms of estimating the entropy/MI, this corresponds to applying a binning with $2^k$ bins to the non-quantized activations as done in the previous studies (Shwartz-Ziv & Tishby, 2017; Saxe et al., 2018), as information available to the network at the time of training is dropped, and thus we expect it to show similar artifacts. Instead, we investigate networks trained using *quantization aware training (QAT)*, in which activations are quantized on the fly during training (Jacob et al., 2018). Specifically, forward passes are quantized while backward passes remain non-quantized. However, the IB and all MI computations in the quantized networks only refer to quantized quantities (i.e., forward pass computations) and thus measure the true information in the network at the considered training step. The actual quantization happens after each layer (after application of the activation function). During training, the algorithm keeps track of the minimum $a_l^{(T)}$ and maximum $a_u^{(T)}$ activations in each layer $T$ across the epoch, and the activation of each neuron in $T$ is mapped to one of $2^k$ uniformly distributed values between $a_l^{(T)}$ and $a_u^{(T)}$, before being passed to the next layer (Jacob et al., 2018). Thus for each epoch, the neurons in a layer $T$ take at most $2^k$ different values, meaning at most $2^{kd_T}$ different states for $T$ (because of the SOFTMAX activation function the effective number of states is $2^{k(d_{\hat{Y}}-1)}$ for the final layer $\hat{Y}$).

## 4 Experiments

We conducted the exact bottleneck analysis in two different setups:

- The setting of Shwartz-Ziv & Tishby (2017) and Saxe et al. (2018) using the network shown in Figure 1a with either TANH or RELU activations, fitted on the same synthetic data set consisting of $|\mathcal{D}| = 2^{12}$ 12-bit binary input patterns with balanced binary output (Section 4.1).
- Learning a two-dimensional representation for the MNIST handwritten digits, where a fully connected network similar to the one in the previous setting using a bottleneck architecture with RELU activations for all hidden layers is trained (Section 4.2).

For each setting, we applied $k = 8$ bit quantization aware training to the given network. As in previous studies (Shwartz-Ziv & Tishby, 2017; Saxe et al., 2018), we used an 80%/20% training/test split to monitor fitting, while computing the MI based on the entire data set (training and test data). The networks were trained using mini batches of size 256 and the Adam optimizer with a learning rate of $10^{-4}$. Weights of layer $T$ were initialized using a truncated normal with mean 0 and standard deviation $1/\sqrt{d_T}$.

In addition, we also replicated the experiments of Shwartz-Ziv & Tishby (2017) and Saxe et al. (2018) using varying number of bins, and we investigated 32- and 4-bit quantization. We also considered IB analysis of a few other architectures for the MNIST task, including architectures with wider bottlenecks, without a bottleneck, and using convolutions. These experiments are described in Supplementary Material A, C, and E.

Our implementation is based on Tensorflow (Abadi et al., 2016). Experiments were run on an Intel Core i9-9900 CPU with 8 3.10GHz cores and a NVIDIA Quadro P2200 GPU.[4].

---

[4]Available at: `https://github.com/StephanLorenzen/ExactIBAnalysisInQNNs`

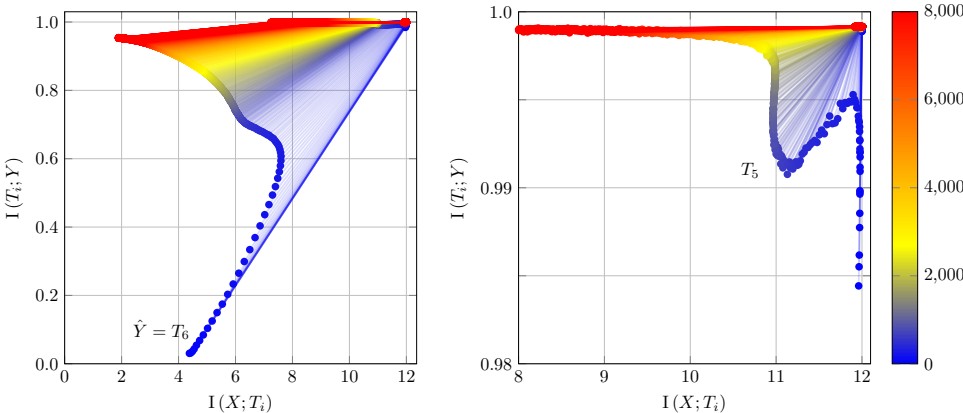

Figure 2: The quantized TANH network, trained in the standard setting with the discrete MI computed exactly. The full information plane (left) and the upper right area (right) are shown. Plotted MI values are the means over the 50 repetitions.

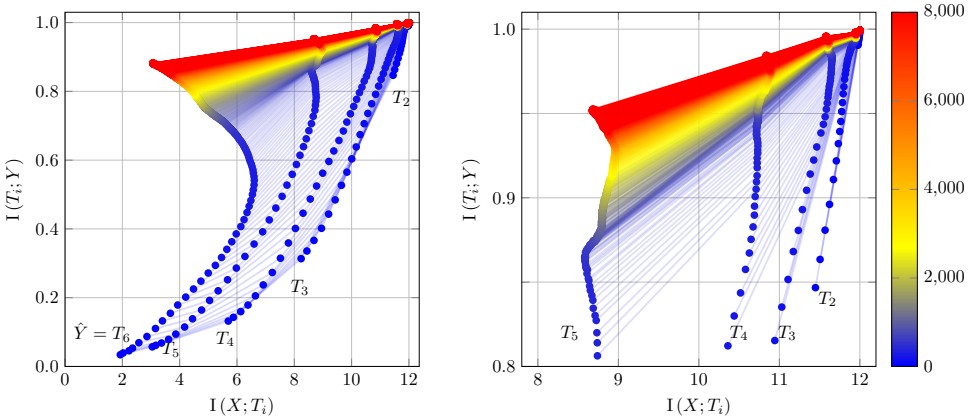

Figure 3: The quantized RELU network, trained in the standard setting with the discrete MI computed exactly. The full information plane (left) and the upper right area (right) are shown. Plotted MI values are the means over the 50 repetitions.

## 4.1 EXACT MUTUAL INFORMATION FOR SYNTHETIC DATA

The 8-bit quantized network from Figure 1a was fitted for 8000 epochs to the synthetic data set considered in Shwartz-Ziv & Tishby (2017), using either TANH or RELU for the hidden layers. The experiment was repeated 50 times and the means of obtained measurements are reported.[5]

Figure 2 and Figure 3 show the information planes for the quantized TANH and RELU networks respectively; additional plots showing the variance of the repetitions are included in Supplementary Material B. Both quantized networks obtained good fits with similar accuracies to non-quantized networks; see Figure F.14 and Figure F.15 in the supplementary material.

For the quantized TANH network, the information plane is very similar to the plane obtained in the standard setting using $2^8 = 256$ bins. We clearly observe a fitting and compression phase in the output layer, while $I(X;T)$ and $I(T;Y)$ in general are large for the input and hidden layers. Looking at the hidden layer before the output layer, we do observe a fitting and a compression phase, but

---

[5]We repeated any trial with an accuracy less than 0.55. This occurred in rare cases, where a RELU network got stuck with a 'dead' layer (all activations are 0 for all inputs) from which it could not recover; such a network is not of interest for the analysis. The risk of 'dead' layers would have been lower if we used a different weight initialization scheme, however, we did not want to change the setup from the original studies.

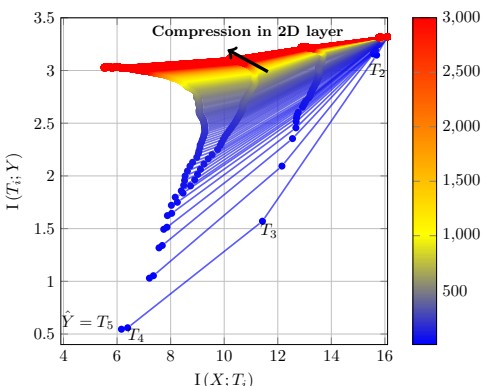

Figure 4: The information plane for the bottleneck structured network fitted on MNIST. Plotted MI values are the means over the 20 repetitions. The compression phase in the 2 neuron layer is highlighted.

with the fitting phase interrupted by a small compression phase. For the remaining layers, the MI measurements with input and label remained almost constant throughout the training process.

Likewise, the information plane for the quantized RELU network looks very similar to the plane obtained with binning. The MI is slightly higher, but the most notably difference is the occurrence of a slight increase in $I(X;T)$ after approximately 1800 epochs, observed only for the hidden layer before the output layer. For the remaining hidden layers, only the fitting phase is observed, with no or very little compression; an observation mirrored in the binning experiments by Saxe et al. (2018).

## 4.2 EXACT MUTUAL INFORMATION FOR LEARNING AN MNIST REPRESENTATION

We considered the task of leaning a two-dimensional representation of the MNIST handwritten digits (Bottou et al., 1994; Deng, 2012). We fitted a network with four hidden layers with RELU activations and 16, 8, 4 and 2 neurons, respectively, where the last two neurons feed into a 10 neuron SOFTMAX output layer. The network was trained for 3000 epochs, and the experiment was repeated 20 times.

The bottleneck architecture for learning a two-dimensional representation leads to a non-trivial IB analysis. Using more expressive networks (e.g., without or with a wider bottleneck) resulted in trivial, constant maximum MI measurements (see Supplementary Material E).

Figure 4 shows the information plane obtained for the network; additional plots showing the variance between repetitions are included in Supplementary Material B. The network exhibits sub-par performance due to the limiting bottleneck structure, obtaining a maximum accuracy of around 90% (see Figure F.16 in the supplementary material). However, the architecture does admit a non-trivial analysis with a clear fitting phase in the output layer and the three last hidden layers. Small compression phases are also observed in the RELU layers around 1500 epochs, similar to the last hidden layer of the network trained on synthetic data. The compression indicates that the learnt representations for digits from the same class become more similar.

## 4.3 DISCUSSION

Our experiments largely confirm the findings by Saxe et al. (2018): we observe no compression phase in the quantized RELU network for the synthetic data. While the quantized networks do not directly correspond to the networks in the original studies (Shwartz-Ziv & Tishby, 2017; Saxe et al., 2018), they amplify the effect of the networks being executed on a digital computer. Note that using quantization (as often done for mobile and edge computing as well as when working on encrypted data) does not violate any assumption in the IB theory – on the contrary, as argued before, it prevents the information theoretic measures from becoming trivial.

The MI generally being higher in our experiments confirms our discussion from Section 3; information is well-preserved through the network. There is no longer a loss of information from the

estimation method. All information loss – and hence the dynamics observed – is a result of the neural network learning. Performing the same experiments with the full 32-bit non-quantized networks, this is even more pronounced, as almost no loss of information is observed between layers. This setting approaches the continuous case, in which we would get $H(T) = \log |\mathcal{D}|$ for invertible activation functions. We include information planes for 32-bit networks with exact MI computation in Supplementary Material C, Figure C.11. Conversely, when training networks with even lower precision, we see a higher information loss between layers. This can be expected from $H(T) \leq 2^{d_T k}$ and the small, decreasing number $d_T$ of neurons in layer $T$ (see Figure 1b). Using $k = 4$ bit precision, more layers are clearly distinguishable, with results similar to those obtained by estimating MI by binning; however, the 4-bit quantized networks also provide a worse fit – indicating that when estimating MI by binning, information vital to the network is neglected in the analysis. We provide information planes for 4-bit quantized networks in Supplementary Material C, Figure C.10.

A similar observation could be made for the network fitted on MNIST. Using the low complexity bottleneck architecture, we obtained non-trivial information measurements. Having only two neurons in the smallest layer limits the network in terms of performance, yielding a sub-optimal accuracy around 90%. Using more complex networks, for instance, with a wider bottleneck or without a bottleneck, we obtained accuracies over 95% (see Figure F.16), but, even when using 8-bit quantization, the information planes were trivial, with only the last layer discernible (see Supplementary Material E). We observed similar trivial information planes for a convolutional architecture, further supporting that only very little information is actually lost in the complex networks (even with quantization), making the information bottleneck, when computed exactly, a limited tool for analysing deep learning.

## 5    CONCLUSIONS AND LIMITATIONS

The experiments and discussion of Shwartz-Ziv & Tishby (2017) and Saxe et al. (2018), as well as several follow-up studies, indicate that *estimating MI by binning may lead to strong artifacts.* We empirically confirmed these issues (see Supplementary Material A): varying the number of bins in the discretization procedure, we observed large differences in the qualitative analysis of the same network, confirming that the results of previous studies highly depend on the estimation method.

To alleviate these issues, we computed the exact MI in quantized neural networks. In our experiments using 8-bit quantized networks, the fitting phase was observed for TANH and RELU network architectures. *The compression phase was observed in the output* SOFTMAX *layer in both cases.* For the TANH network, the MI was only non-trivial for the last two layers, but not for the other hidden layers. *We observed compression in the last hidden layer when using* TANH *activations*; similar to the phase observed by Shwartz-Ziv & Tishby (2017). For the RELU network, the MI was below its upper bound in all layers; however, *no compression was observed in any of the hidden layers using* RELU *activations*. The latter observation confirms the statements by Saxe et al. (2018) about the IB in RELU networks.

Applying the analysis to a network trained on MNIST, we found that a bottleneck architecture learning a low-dimensional representation showed a compression phase. The compression indicates that the learnt representations for digits from the same class become more similar, as intended for representation learning. When applied to networks without a bottleneck (e.g., a simple convolutional architecture), the IB analysis produced only trivial information planes, which shows the limitations of the IB analysis (even with exact computations) for deep learning in practice.

The main contribution of our work is that *we eliminated measurements artifacts from experiments studying the IB in neural networks*. By doing so, future research on the IB concept for deep learning can focus on discussing the utility of the IB principle for understanding learning dynamics – instead of arguing about measurement errors.

The proposed quantization as such is no conceptual limitation, because for invertible activation functions studying a (hypothetical) truly continuous system would lead to trivial results. The biggest limitation of our study is obviously that *we could not (yet) resolve the controversy about the compression phase* (which was, admittedly, our goal), because our experiments confirmed that it depends on the network architecture whether the phase is observed or not. However, the proposed framework now allows for accurate future studies of this and other IB phenomena.

## REPRODUCIBILITY STATEMENT

Source code and instructions for reproducing our experiments are available online: `https://github.com/StephanLorenzen/ExactIBAnalysisInQNNs`

## ACKNOWLEDGEMENTS

The authors would like to thank the anonymous reviewers for their constructive comments. SSL acknowledges funding by the Danish Ministry of Education and Science, Digital Pilot Hub and Skylab Digital. MN acknowledges support through grant NNF20SA0063138 from Novo Nordisk Fonden. CI acknowledges support by the Villum Foundation through the project Deep Learning and Remote Sensing for Unlocking Global Ecosystem Resource Dynamics (DeReEco). The authors acknowledge support by the Pioneer Centre for AI (The Danish National Research Foundation, grant no. P1).

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

## A    ESTIMATING MUTUAL INFORMATION BY BINNING

This section presents a replication of the experiments from Shwartz-Ziv & Tishby (2017) and Saxe et al. (2018), but varying the number of bins for estimating MI. Non-quantized TANH and RELU networks were trained in the standard setting, and MI was estimated using 30, 100 and 256 bins. The number of bins were chosen to match the numbers used in the earlier works (30 used by Shwartz-Ziv & Tishby, 100 used by Saxe et al.) and to match the discretization in the quantized neural networks considered in Section 4.1.

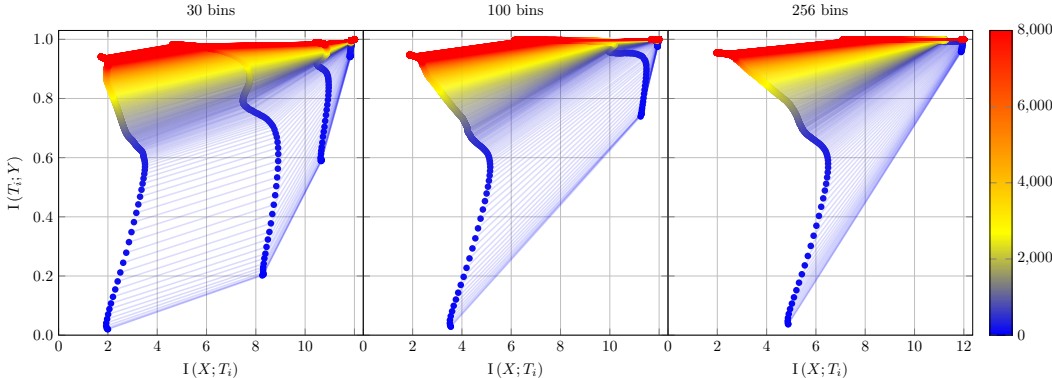

Figure A.5: Information planes for the TANH network with various number of bins. 30 bins corresponds to a replication of the results from Shwartz-Ziv & Tishby (2017). The network is trained in the standard setting, and the average of 50 runs is plotted.

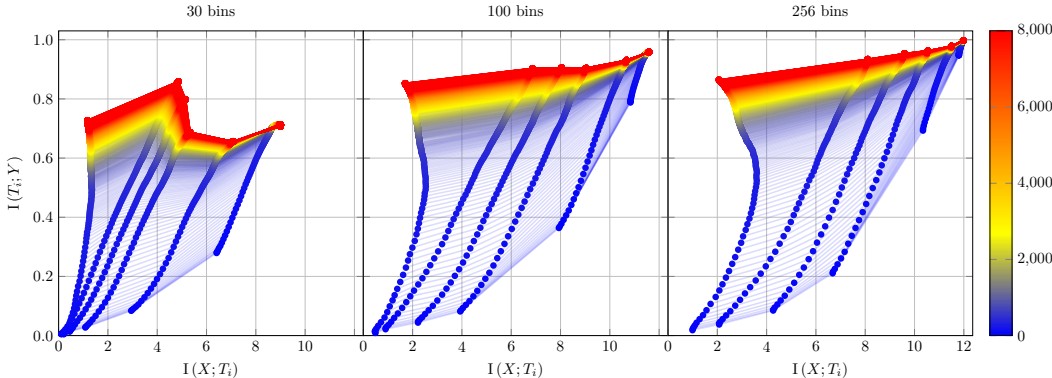

Figure A.6: Information planes for the RELU network with various number of bins. 100 bins replicate the results from Saxe et al. (2018). The network is trained in the standard setting, and the average of 50 runs is plotted.

The resulting information planes are shown for the TANH and RELU networks in Figure A.5 and Figure A.6 respectively.

From Figure A.5, we clearly see the results of Shwartz-Ziv & Tishby (2017) replicated using 30 bins for the TANH network; two phases are clearly visible in each layer. As expected, for every layer $T$ with binned $T' = B(T)$, $I(X; T')$ and $I(T'; Y)$ increase with the number of bins used with the phases still remaining visible in the distinguishable layers.

For the RELU network with 100 bins, the results of Saxe et al. (2018) were also replicated, that is, the compression phase was not observed. When using only 30 bins for the RELU network, the estimation broke down completely in the sense that the DPI (2) is violated: $I(T'; Y)$ is non-monotone, a phenomenon occurring because of the estimation, which has also been observed by Geiger (2020). Again, we see for any layer $T$ that $I(X; T')$ and $I(T'; Y)$ increase with $m$, although to a lesser degree

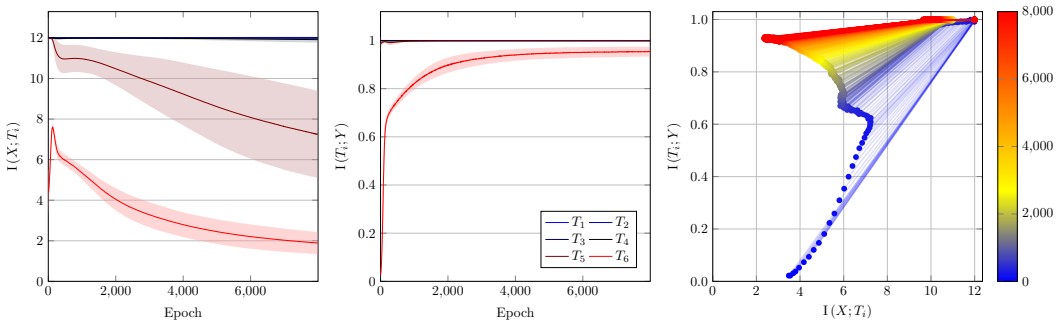

Figure B.7: Mean and variance for $\mathrm{I}\left(X;T\right)$ (left) and $\mathrm{I}\left(T;Y\right)$ (center), and the information plane for the median deviating (L2-distance from the mean) repetition (right), for the TANH network in the standard setting.

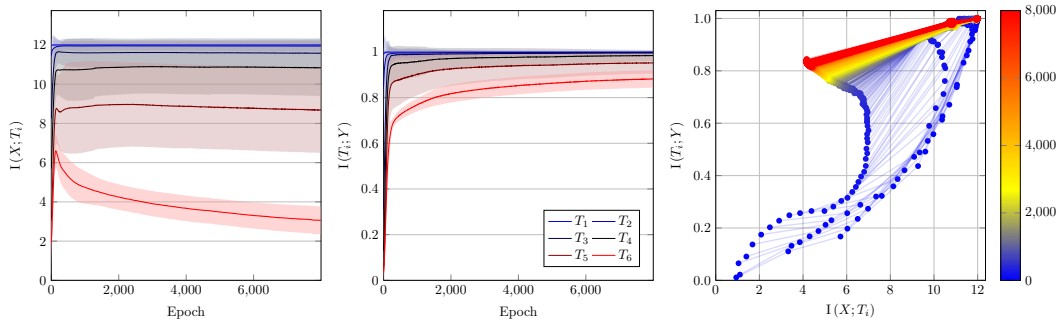

Figure B.8: Mean and variance for $\mathrm{I}\left(X;T\right)$ (left) and $\mathrm{I}\left(T;Y\right)$ (center), and the information plane for the median deviating (L2-distance from the mean) repetition (right), for the RELU network in the standard setting.

than for the TANH network, most likely due to the RELU activation functions actually dropping information (by zeroing negative activations).

In general, the estimation of MI in either network is highly dependable on the number of bins used, indicating that the interpretation of the information plane is not straight forward, when using binning for estimation.

# B  MI VARIANCE

In this work, we investigate the effect of MI approximation in previous IB studies. These original studies are based on a qualitative analysis of mean curves. However, the question arises of whether mean curves correctly represent individual learning processes. While we do not investigate this question in detail, the following section provides plots illustrating the variance between experimental trials.

Figures B.7, B.8 and B.9 show the mean and variance of $\mathrm{I}\left(X;T\right)$ (left) and $\mathrm{I}\left(T;Y\right)$ (center), as well as the information plane for a single trial (right), for the TANH and RELU networks in the standard setting, and the bottleneck network applied to MNIST, respectively. For the single trial, we selected the median deviating repetition, ranked by the L2-distance from the mean.

From the plots, we see that when the mean MI is large, the variance is usually low. The variance is similar across the different networks (note the different scaling of the y-axes), with slightly larger variance for the RELU activations. Together with the median deviating information plane, the plots suggest that the mean information plane, which averages out less pronounced phases of MI increase and decrease, represents individual repetitions fairly well.

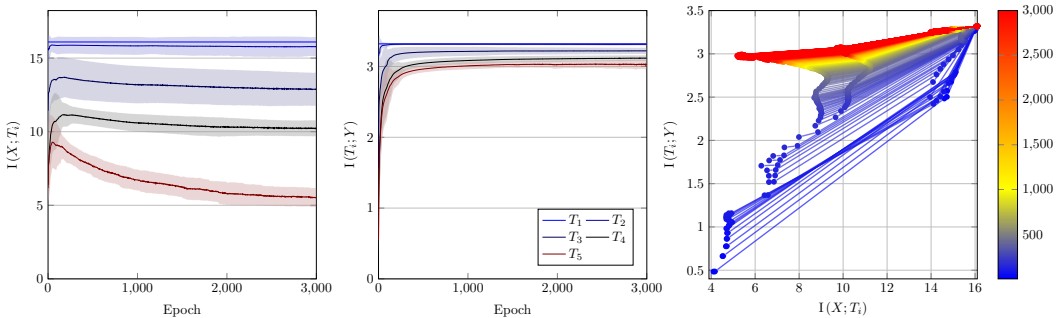

Figure B.9: Mean and variance for $I(X;T)$ (left) and $I(T;Y)$ (center), and the information plane for the median deviating (L2-distance from the mean) repetition (right), for the Bottleneck network applied to MNIST.

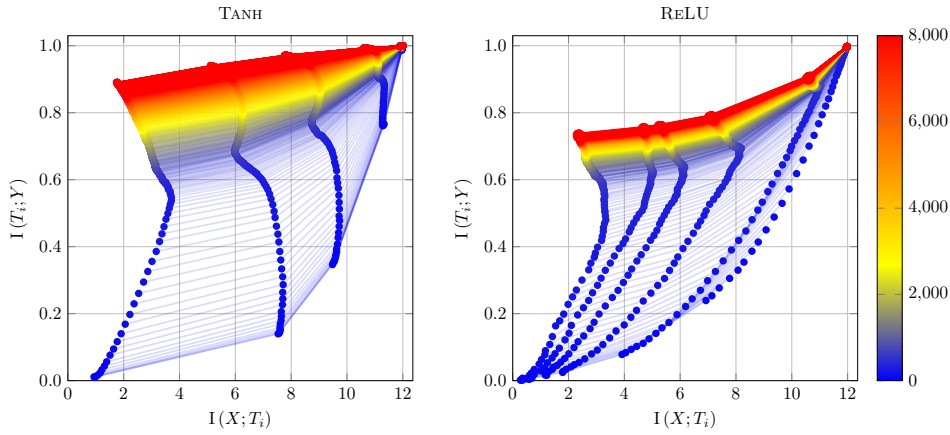

Figure C.10: Information planes for 4-bit quantized TANH network (left) and RELU network (right). The mean of 30 runs is plotted.

Additionally, the compression phase in the TANH network can clearly be seen in Figure B.7 (left), while it is absent in the RELU network (Figure B.8).

## C  EFFECT OF THE QUANTIZATION PRECISION

This section presents information planes obtained using 4- and 32-bit quantized neural networks trained in the standard setting, in order to investigate the effect of the precision used in the quantization.

Figure C.10 depicts the resulting information planes for the 4-bit networks. As each neuron can take only $2^4 = 16$ different values, the total number of possible states per layer decreases significantly in this setting, and as expected we see lower overall MI measurements. For the TANH network, several more layers become distinguishable. The observed information planes looks similar to those observed in the original experiments by Shwartz-Ziv & Tishby (2017) and Saxe et al. (2018). However, the network accuracy has now degraded compared to the non-quantized networks (see Supplementary Material F), which indicates that the binning used in the estimation of the MI in previous experiments has discarded information vital to the network.

Figure C.11 shows the resulting information planes for the 32-bit networks. As expected, we see an overall increase in MI; the information drops only very slowly through the network. Each layer has many possible states and – given the small data set – we we get closer to the behavior of a continuous system.

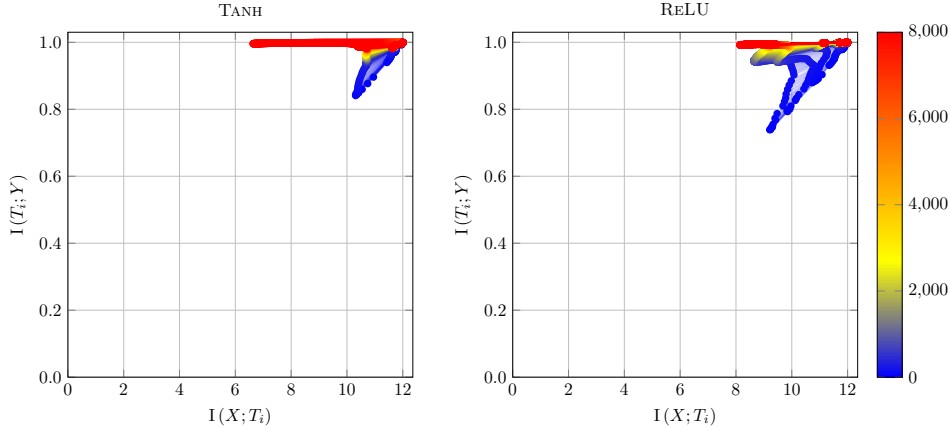

Figure C.11: Information planes for 32-bit quantized TANH network (left) and RELU network (right). The mean of 30 runs is plotted.

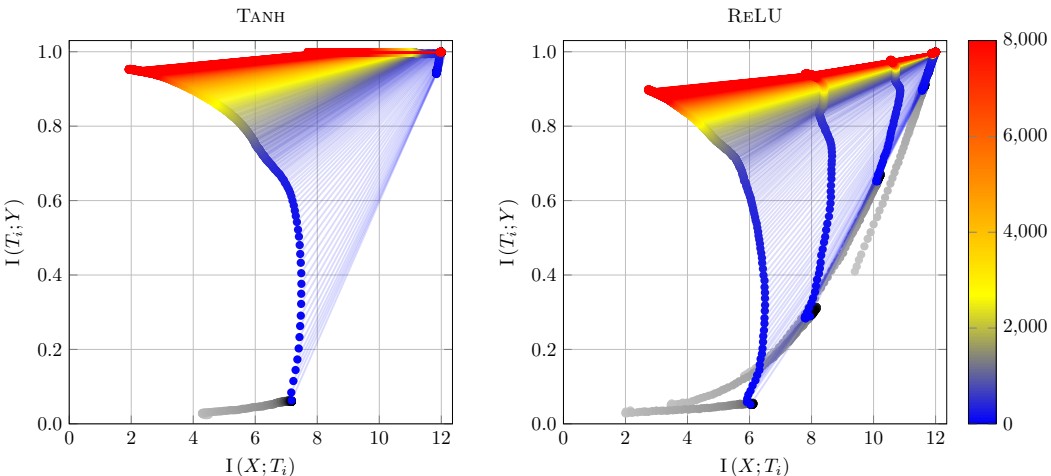

Figure D.12: The information planes for the TANH (left) and RELU (right) networks when prefitting on random labels for 1000 epochs. The means of 20 repetitions are shown. In the plots, $Y$ always refers to the true labels.

## D  QUANTIZED NETWORK WITH RANDOMIZED PREFITTING

This section presents results for the standard setting using 8-bit quantized networks, but with weights prefitted to random labels. First, the network is fitted for 1000 epochs to the synthetic data set with the labels shuffled. The network is then trained for 8000 epochs with correct labels (as done in Section 4.1). The experiment is repeated 20 times. The resulting information planes for the TANH and the RELU network are presented in Figure D.12, left and right respectively.

Inspecting the plots, we see that, for all discernible layers, $I(X;T)$ appears to increase when fitted to random labels. This is not surprising as the input is not shuffled.

As a consequence of the increase in $I(X;T)$, the hidden layers (most notable in the RELU network) also see an increase in $I(T;Y)$. As the ability to distinguish different inputs from a hidden layer $T$ increases, so does $I(T;Y)$.

As expected, the network accuracy does not increase when training on random labels, and accordingly there is no lateral movement in the information plane for the output layer.

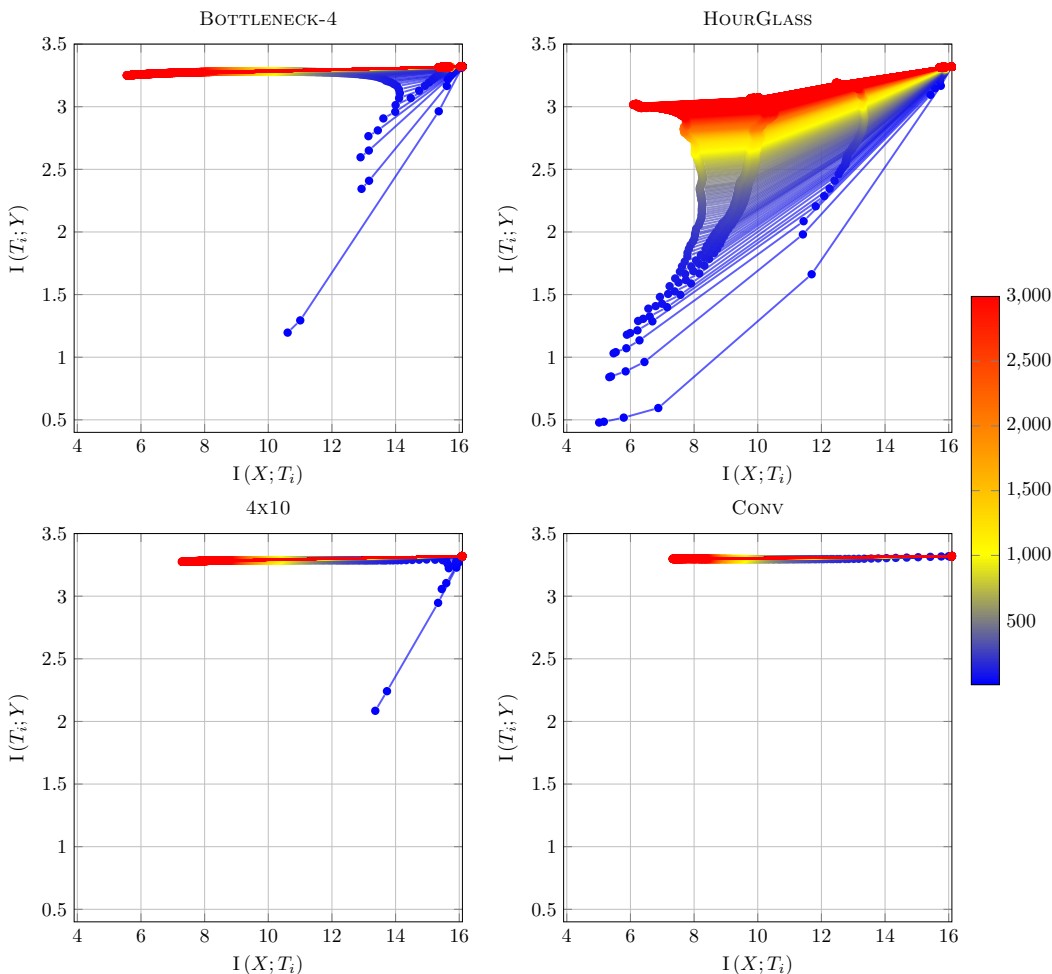

Figure E.13: Information planes for the additional networks trained on MNIST. Note for HOURGLASS that the three last hidden layers almost coincide completely.

## E   MNIST WITH OTHER NETWORK ARCHITECTURES

This section presents the results of training a number of 8-bit quantized networks with varying architectures on MNIST. We consider the same setup as in Section 4.2, but with the following architectures (all using RELU activations in the hidden layers and a final 10 layer SOFTMAX output):

- The BOTTLENECK-4 network with 4 hidden layers of widths 16, 12, 8 and 4, i.e. the bottleneck has width 4.
- The HOURGLASS network with 6 hidden layers of widths 16, 8, 4, 2, 4, 8. The bottleneck width is still 2, but the network expands again after the bottleneck, creating an hourglass shape.
- The 4X10 network with 4 hidden layers, each of width 10.
- The CONV network, a simple convolutional network with structure: CV-MP-CV-MP-FC, where CV denotes a (3,3) 2-channel convolutional layer, MP denotes a (2,2) max pooling and FC denotes a fully connected RELU layer with 20 neurons.

Below, we denote the network used in Section 4.2 by BOTTLENECK-2. Each experiment is repeated 20 times and the resulting information planes are reported in Figure E.13. As expected, given the increased complexity, all networks exhibit better performance than the bottleneck structured network considered in Section 4.2 (the accuracy of all networks are presented in Figure F.16 in Supplementary

Material F), obtaining accuracies close to or above 95% for all networks except the HOURGLASS network.

From Figure E.13, we observe the information curves for the BOTTLENECK-4 network to be similar in shape to the BOTTLENECK-2, but with higher MI measurements in general. This is not surprising, as the layers are wider and thus more expressive.

As expected, the HOURGLASS network also exhibits information curves similar to BOTTLENECK-2 (as the architecture is similar for the first few layers). Not surprisingly, the two expanding layers of widths 4 and 8 almost coincide completely with the bottleneck layer; information cannot increase but is also not decreased significantly.

The two more expressive networks without bottlenecks, 4X10 and CONV, have almost completely trivial information curves. Limited fitting for the output layer in 4X10 and compression in the output layers for both networks can be observed. The experiments indicate the limitations of exact IB analysis for complex, large scale networks.

## F  NETWORK ACCURACIES

Figure F.14 and Figure F.15 report the accuracies obtained by the TANH and RELU networks when fitted on the synthetic data, respectively. As can be seen, the networks suffers only slightly when

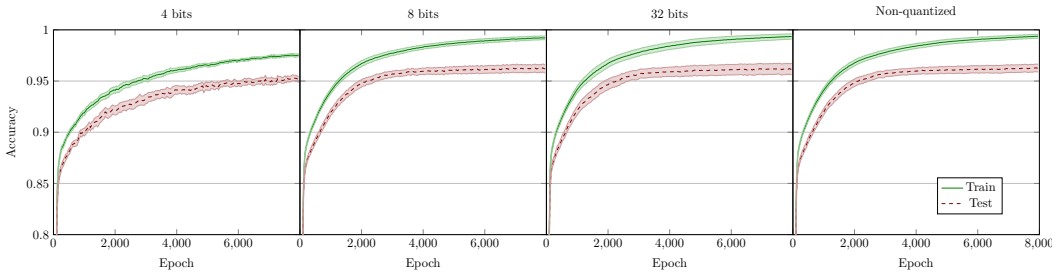

Figure F.14: Accuracies of the quantized TANH networks (three on the left) and the non-quantized TANH network (right). The means and 95% confidence intervals over 50 repetitions are reported.

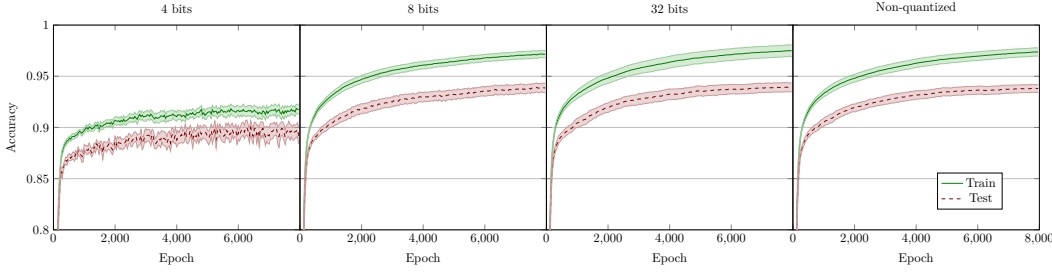

Figure F.15: Accuracies of the quantized RELU networks (three on the left) and the non-quantized RELU network (right). The means and 95% confidence intervals over 50 repetitions are reported.

using only 4 bits in the quantization.

Figure F.16 reports the accuracy of the networks fitted to MNIST. Unsurprisingly, the more complex networks (BOTTLENECK-4, 4X10, CONV) obtain better accuracy with lower variance, compared to the networks with a low-width bottleneck (BOTTLENECK-2, HOURGLASS).

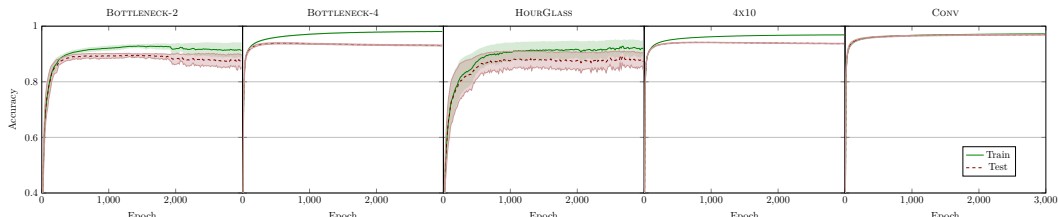

Figure F.16: Accuracies of the quantized networks applied to MNIST. The means and 95% confidence intervals over 20 repetitions are reported. The decrease in training accuracy for the bottleneck structured network (left) is due to the optimization objective being cross-entropy.

