# OpenReview forum: "Information Bottleneck: Exact Analysis of (Quantized) Neural Networks"
_ICLR.cc/2022/Conference — ICLR 2022 Poster_

### Official Review · Reviewer_KkRc · 2021-11-01

**Correctness:** 3
**Technical Novelty And Significance:** 2
**Empirical Novelty And Significance:** 3
**Recommendation:** 6
**Confidence:** 4

**Main Review:**

**Quality, Clarity, Soundness, Correctness**

The paper is very well written, and the necessary background including the main controversy and claims is well presented (perhaps a bit too detailed even regarding the history of the controversy). Methods and results are clear, and important ablations are included in the appendix. My main criticism is that the results qualitatively depend on the quantization bit-width chosen. I greatly appreciate that the authors do not “shove this under the carpet” but discuss it in the main paper and show it in the appendix. This will be valuable to anyone looking to implement the same main idea. However, in terms of take-aways from the paper the main claims and the controversy remain elusive. While it is undeniable that the paper analyzes training dynamics of activation-quantized networks exactly - it is unclear how these dynamics relate to the training dynamics of “unquantized” networks analyzed via binning. More importantly, there is no way of objectively choosing the right quantization width (which is perhaps similar to not being able to choose the right (non-uniform) binning). This could leave different people drawing different conclusions while using the same method (somewhat similar to the original controversy). Therefore, I would not call the results or method incorrect - but I also would not claim that this paper solves the controversy once and for all.


**Verdict**

I enjoyed reading this paper - it is well written, presents the background and history well and adds another piece to the continuing saga. The main idea is straightforward but good, and experiments and ablations are well executed. Unfortunately, the paper does not conclude the saga. My two main points of criticism are:
1) The paper can only say something about training dynamics of activation-quantized networks - and while it is not far-fetched to assume that they relate well to analyzing training dynamics of “unquantized” networks analyzed via simple binning, this relationship is not well understood at all (in fact it is conceivable that the regularization effect of the strong quantization interacts with the compression effect of SGD).
2) Perhaps the main worry is that the results qualitatively depend on the quantization bit-width chosen. This again leaves room for different interpretations of the findings which the method cannot objectively resolve.
Overall I think there is merit to publishing this paper since it is well executed and might inform other people attempting to follow the same main idea. I am currently leaning slightly towards acceptance; my only reservation would be the somewhat limited significance of the paper (given the goals and the two main problems mentioned above).

SideNote: Generally speaking I think it might be time to evaluate what information-plane analysis really tells us about training dynamics and how to use these findings. Perhaps the main merit is to provide inspiration for theoretical analysis of SGD dynamics, rather than empirically measuring the strength of a “compression phase”. Ideally these results would lead to a degree of understanding where we know which factors influence the fitting and compression phase, and how this influence plays out exactly - and perhaps most importantly how SGD can have a regularizing effect that helps with generalization. Some works have already set out towards this path. Another line of attack could be through a series of control experiments where information-quantities are known (and estimation methods can be calibrated/evaluated against), and where training-dynamics can be controlled (e.g. there should be no non-trivial compression phase when training with random labels; if we initialize with a solution that has “memorized” the entire training-set, do we see a greatly accelerated fitting phase? does compression have an immediate onset, how does it depend on hyper-parameters, network-capacity, activation functions, etc.?).

**Improvements**

1) Even though difficult, the paper would benefit strongly from an objective criterion on how to choose the quantization bit-width, or a “calibration” process to find such a value. I know this is a long shot - but to me this would be the most significant improvement of the current work.

2) Though a uniform quantization is the most simple, is it well justified? Empirically activation distributions are far from being uniform between the min and max. How would a non-uniform quantization scheme change results?

3) Control experiment (just a suggestion, I do not expect to see results in the authors’ response): train the network to fit random labels and use this as an initialization rather than random initialization of weights. What do the information dynamics look like when training (with correct labels) from this initialization? Do we see strongly accelerated fitting, do we get immediate compression? Do we see any compression in the networks/layers where we typically don’t see a compression phase?

**Detailed comments**

1) Would be nice to add one/two sentences on how gradients are backpropagated through the quantization functions (or are these simply bypassed for the backwards pass?).

2) Fig 2, 3 - please indicate which layer is which in the plots.

3) Maybe shorten/move the historical bit (openreview comments) to a footnote. I really like the very comprehensive list of references and their findings.

4) Page 2 first para: maybe emphasize that IB/rate-distortion are lossy compression.

5) Page 2 after first para: would be helpful for the reader to write down the IB objective.

6) Page 2 second para: sounds a bit like the DPI needs to be carefully taken into account, but it cannot be violated. Maybe rephrase to: “The DPI of course holds in the IB, …”.

 7) Page 6 second para: typo “quantized after training”.


**Summary Of The Paper:**

**Update after authors' response:** The authors have managed to clarify some issues and make a couple of small improvements to the manuscript. I am tempted to raise my score to a 7, but to me personally the paper does not quite pass the threshold for an 8 (which is the next possible rating on the conference scale). I am in favor of accepting the paper and will argue so during the reviewers' discussion.

**Summary**

The paper investigates the information-plane analysis of deep neural network training dynamics. The original claim was that neural networks go through distinct fitting and compression phases in SGD training that are well characterized by the information plane. These findings have later been disputed (particularly due to the way mutual information is estimated) which has lead to tens of papers investigating the phenomenon with mixed conclusions. The main idea of this paper is to eliminate the estimation problem by training neural networks with quantized activations, where the discrete mutual information can be computed exactly (in the limit of infinite samples). The original claims, controversy, and follow-up works are introduced and discussed in great detail. Original experiments are repeated with a very similar protocol to facilitate comparability of results. Additional experiments on MNIST are conducted, and ablations are performed. Results show the two distinct phases for some activation functions and some layers, but not for other activation functions.

**Main contributions**

1) Thorough discussion of the history of the Information-plane analysis, the main claims and previous findings and the follow-up papers it has spawned. Significance: The background and related work is well researched and well presented, which is very helpful for readers who have not followed this line of research closely. The only downside is that there is a fairly recent review (which is cited in the paper) which somewhat limits significance.

2) Quantized-activation training to avoid estimation errors when computing mutual information terms. Significance: the idea is very sensible in principle. Unfortunately, as some of the ablations show, the qualitative results can depend strongly on the quantization bit-width chosen. This is unfortunate since the main goal was to eliminate the influence of binning in naive estimation of the mutual information. Nonetheless, the results are exact and reliable for training quantized-activation neural networks - what is not clear is how to choose the quantization.
3) Reproduction of original experiments of Schwartz-Ziv & Tishby and Saxe et al. By sticking as closely as possible to the original training protocols, the new results are as comparable as possible (which does not mean that it is guaranteed that quantized-activation training dynamics are similar to “non-quantized” dynamics analyzed via binning - but empirically this seems to hold to a large degree). The paper also shows interesting ablations and additional experiments on MNIST. Significance: The most important experiments to run are included in the paper and important ablations are shown. The significance of the results could be improved by showing larger scale experiments on different kinds of networks (CNNs, ResNets, Transformers, …).


**Summary Of The Review:**

The paper nicely summarizes some recent findings regarding information-plane analysis of neural network training, the dispute that followed the original results, and further follow-up work. To resolve some of the main issues (stemming from potential estimation errors of mutual information) the paper proposes to use quantized-activation networks instead, where the mutual information terms can be computed exactly. This idea is simple, yet compelling - unfortunately the problem of how to choose the correct quantization bit-width remains. Crucially, different bit-widths can lead to qualitatively quite different conclusions and there is currently no objective way of choosing the bit-width. The paper shows this via ablations which are discussed in the main text. Additionally, it is currently unclear how the training dynamics of (heavily) quantized networks relate to the training dynamics of “non-quantized” networks; after all; the activation-quantization could potentially have a regularizing effect that might interact with the compression induced by SGD alone. I think the paper is executed well, which is why I am currently in favor of sharing the results of this fairly straightforward idea with the wider community. I am not sure though whether this approach in general is a fruitful direction to target further research at in order to understand SGD training dynamics - which limits the potential impact of this paper.

---

### Official Review · Reviewer_bFPZ · 2021-11-02

**Correctness:** 4
**Technical Novelty And Significance:** 2
**Empirical Novelty And Significance:** 4
**Recommendation:** 8
**Confidence:** 3

**Main Review:**

I find the paper very well written and interesting. It is non-technical and purely empirical, but the numerical experiments are well-carried and insightful, on a difficult and timely subject. The authors DO mention that they did not provide an answer to the motivating question at the root of the work, namely, on whether compression phases happen in general in neural network training, because their experiments show that it depends on the setting and activation function used (it may also be that compression happens or not truly depending on the setting, and so there is not a definite answer). But it brings on the table a scheme to approach that question more systematically and cleanly. I believe that the paper is a very valuable contribution and will motivate other groups to take their approach. I do not see any obvious weakness other than there is no theory, but it is completely fine for a well-done empirical paper.

**Summary Of The Paper:**

This paper considers the important problem of mutual information estimation in neural networks, a problem at the root of a debate on the usefulness of the information-bottleneck approach for the analysis of information flow in neural networks. There exist many approximation schemes to get estimates of the inter-layers mutual information, but the issue is that they may lead to different conclusions due to their sensitivity to discretization schemes. The authors propose instead to study discretized neural nets, trained with a simple learning procedure taking into account the discretization and that thus does not need post-training discretization. In this case mutual informations can be computed exactly and the data quantifies the true information flow along training.

**Summary Of The Review:**

I recommend publication, as the paper is well-witten and clear, and brings about a new scheme and neural net model to compute exaclty one of the most fundamental quantity in their training: the mutual information.

---

### Official Review · Reviewer_vYX6 · 2021-11-03

**Correctness:** 3
**Technical Novelty And Significance:** 2
**Empirical Novelty And Significance:** 2
**Recommendation:** 6
**Confidence:** 5

**Main Review:**

UPDATE: Reading the revised manuscript, I am willing to improve my rating. The paper has merits and deserves to be published, even though I have still reservations regarding a few aspects. See reply to authors' response for details.

The paper is interesting, well written, and covers a relevant controversy (albeit in a niche). While previous works focused predominantly on deterministic networks with continuous activation functions (in which the estimation of information quantities is problematic), this work is among the first in which the information planes (IPs) can be interpreted from an information-theoretic perspective. Despite this merit, the technical contribution of this paper seems to be limited by the considerations below:

- Methodologically, the paper is indeed similar to Raj et al. 2020. The authors claim that their approach is still different because Raj et al. 2020 use different training schemes (such as the straight-through estimator) and that thus the training trajectories will be fundamentally different. Does this not also affect the quantization-aware training that was used here? If so, what insights can be transferred to vanilla SGD?
- The authors claim that the resulting mutual information values are "true", i.e., that the effects of mutual information estimation are entirely removed. This statements needs qualification. Indeed, if the empirical distribution of the dataset D is considered the true distribution P_XY, then this statement is correct. However, if we assume that there is a true distribution P_XY from which D is sampled iid (which is a more common assumption), then the computed mutual information values are indeed again only estimates, estimated from a finite dataset D. I suggest to at least make the former assumption explicit (and to defend it accordingly).
- Furthermore, the authors present IPs by averaging over 50 training trajectories. This may be problematic if the phenomena are not epoch-aligned over these 50 runs. Indeed, averaging may explain certain "dents" in the IP that are observed in, e.g., Figs. 3, A.5., and C.8. For the sake of completeness, I suggest to show (maybe in the supplementary material) overlaid IPs of individual layers for all 50 runs to see if they fall on top of each other or not.
- Regarding the generality of the paper's claims and connecting to the authors statement that the presence of a compression phase depends on the network architecture: The manuscript would be stronger if the authors add several more network architectures to investigate the effects, e.g., by varying the bottleneck size of the bottleneck architecture for MNIST, but also considering entirely different architectural choices. That the presence of compression depends on the architecture is not a problem -- the question how it depends on the architecture remains interesting and may deserve further research.
- Finally, it is not fully clear what practical or theoretical implications the results of this paper have. The controversy about the existence of a compression phase is known. Several authors have hinted at a possible connection between such a compression in the IP and geometric effects in latent space (e.g., see Goldfeld et al. 2019 or Geiger 2020). Whether there is a connection between compression in the IP and generalization is unclear, but Saxe et al. 2017 and Goldfeld et al. 2019 suggest that there is not. How does the present manuscript complement these insights?
- Also, there seems to be little connection between the learning curves in Appendix E and phenomena observed in the IPs. Specifically, while there seems to be compression for tanh activation functions but not for ReLU, the corresponding learning curves do not look qualitatively differently. What conclusions can we draw from that?

**Summary Of The Paper:**

The paper discusses the information plane of quantized neural networks. The authors further investigate whether or not there are compression phases during training, a question over which there has been much controversy in the past.

**Summary Of The Review:**

An interesting, although small, contribution to a recent controversy in understanding deep neural networks.

---

### Comment · Reviewer_bFPZ · 2021-11-24
**Update post authors feedback**

Dear all,

After reading the other reviews and feedback, I decided to maintain my score to 8, but lower my confidence from 3 to 2 as I have hard time to gauge the originality of the approach. I still believe that the paper is a great contribution and well written on a timely problem, thus my score. But I did not notice a number of points raised by other reviewers, and I clearly lack knowledge on the literature that some other more confident reviewers seem to have. Overall, it seems anyway that the consensus is pretty positive apart from vYX6. I'm curious to know about her/his updated opinion.

---

### Decision · Program_Chairs · 2022-01-20

**Decision:**

Accept (Poster)

**Comment:**

Initially, some reviewers have raised several points of criticism regarding certain aspects of the model whose novelty/significance was a bit unclear. After the rebuttal and the discussion phase, however, everyone agreed that most of these concerns could be addressed in a convincing way, and finally all reviewers were in favor of this paper. After carefully going over all the reviews, the rebuttal and the discussions, I fully agree with the reviewers and came to the conclusion that this paper indeed contains some interesting, novel and relevant contributions.